# Vegetation distribution and terrestrial carbon cycle in a carbon-cycle configuration of JULES4.6 with new plant functional types

Anna B. Harper[1], Andrew J. Wiltshire[2], Peter M. Cox[1], Pierre Friedlingstein[1], Chris D. Jones[2], Lina M. Mercado[3,4], Stephen Sitch[3], Karina Williams[2], Carolina Duran-Rojas[1]

1: College of Engineering, Mathematics, and Physical Sciences, University of Exeter, Exeter EX4 4QF, U.K.

2: Met Office Hadley Centre, Fitzroy Road, Exeter EX1 3PB, U.K.

3: College of Life and Environmental Sciences, University of Exeter, Exeter EX4 4PS, U.K.

4: Centre for Ecology and Hydrology, Wallingford OX10 8BB, U.K.

*Correspondence to:* Anna B. Harper (a.harper@exeter.ac.uk)

**Abstract.** Dynamic global vegetation models (DGVMs) are used for studying historical and future changes to vegetation and the terrestrial carbon cycle. JULES (the Joint UK Land Environment Simulator) represents the land surface in the Hadley Centre climate models and in the UK Earth System Model. Recently the number of plant functional types (PFTs) in JULES were expanded from 5 to 9 to better represent functional diversity in global ecosystems. Here we introduce a more mechanistic representation of vegetation dynamics in TRIFFID, the dynamic vegetation component of JULES, that allows for any number of PFTs to compete based solely on their height, removing the previous hardwired dominance hierarchy where dominant types are assumed to outcompete subdominant types.

With the new set of 9 PFTs, JULES is able to more accurately reproduce global vegetation distribution compared to the former 5 PFT version. Improvements include the coverage of trees

within tropical and boreal forests, and a reduction in shrubs, which dominated at high latitudes. We show that JULES is able to realistically represent several aspects of the global carbon cycle. The simulated gross primary productivity (GPP) is within the range of observations, but simulated net primary productivity (NPP) is slightly too high. GPP in JULES from 1982-2011 is 133 PgC yr$^{-1}$, compared to observation-based estimates between 123±8 (over the same time period) and 150-175 PgC yr$^{-1}$. NPP from 2000-2013 is 72 PgC yr$^{-1}$, compared to satellite-derived NPP of 55 PgC yr$^{-1}$ over the same period and independent estimates of 56.2±14.3 PgC yr$^{-1}$. The simulated carbon stored in vegetation is 542 PgC, compared to an observation-based range of 400-600 PgC. Soil carbon is much lower (1422 PgC) than estimates from measurements (>2400 PgC), with large underestimations of soil carbon in the tropical and boreal forests.

We also examined some aspects of the historical terrestrial carbon sink as simulated by JULES. Between the 1900s and 2000s, increased atmospheric carbon dioxide levels enhanced vegetation productivity and litter inputs into the soils, while land-use change removed vegetation and reduced soil carbon. The result is a simulated increase in soil carbon of 57 PgC but a decrease in vegetation carbon of 98 PgC. The total simulated loss of soil and vegetation carbon due to land-use change is 138 PgC from 1900-2009, compared to a recent observationally-constrained estimate of 155±50 PgC from 1901-2012. The simulated land carbon sink is 2.0±1.0 PgC yr$^{-1}$ from 2000-2009, in close agreement to estimates from the IPCC and Global Carbon Project.

**1. Introduction**

Dynamic global vegetation models (DGVMs) are used for predicting changes in vegetation distribution and carbon stored in the terrestrial biosphere (Prentice et al., 2007; Fisher et al., 2014). When coupled to climate models, these tools enable the study of interactions between climate change, land use patterns, and the terrestrial carbon cycle. Typically, DGVMs either group the world's vegetation types into plant functional types (PFTs), or aggregate vegetation sharing a

common biogeography into biomes (Woodward, 1987; Running and Gower, 1991; Prentice et al., 1992). A move towards a PFT approach recognized the differential response of plant function to rapid future climate change (Foley et al., 1996; Sitch et al., 2003). However, due to data limitations
these models were handicapped in the number of PFTs they could define and differentiate.

JULES (Best et al., 2011; Clark et al., 2011) is a DGVM that represents the land surface in the UK Hadley Centre family of models (e.g. the UK Earth System Model in the 6[th] phase of the Coupled Model Intercomparison Project, CMIP6, and the HadGEM2 models in CMIP3 and CMIP5). Within
JULES, TRIFFID (Top-down representation of Interaction of Foliage and Flora Including Dynamics; Cox, 2001) predicts changes in the carbon content of vegetation and soils, and vegetation competition. Since its creation in the late 1990's, competition in TRIFFID was limited to between five PFTs (broadleaf trees, needle-leaf trees, C3 and C4 grasses, and shrubs). Under this approach, each PFT competed with other PFTs based on a prescribed hierarchy, where dominant
PFTs were assumed to outcompete subdominant PFTs. The proliferation of new ecological data over the past decade has provided the opportunity to improve TRIFFID and the entire JULES model on a range of scales: for example, the TRY database stores detailed information on plant traits that are important for the processes of photosynthesis and respiration (Harper et al., 2016), while on the global-scale new vegetation maps enable improved analysis of predicted plant distributions (e.g.
(Poulter et al., 2015). Exploitation of these new datasets allow a more detailed representation of vegetation distribution and the terrestrial carbon cycle, and improve the biophysical characterization of the land-surface in climate models (e.g. albedo implications of deciduous versus evergreen phenology in boreal forests).

The physiology of JULES was recently updated to include the following leaf traits: leaf mass per unit area, leaf nitrogen per unit mass, and leaf lifespan. An iterative process of development and evaluation with JULES resulted in an improved representation of gross and net primary productivity

(GPP and NPP, respectively) based on an expanded set of PFTs (Harper et al., 2016). The new PFTs were also used in the development and evaluation of a new fire module in JULES (INteractive Fire and Emission algoRithm for Natural envirOnments, or INFERNO; Mangeon et al., 2016). However, given the primary focus on improved physiology, the Harper et al. (2016) study adopted a prescribed vegetation distribution based on satellite data. Here we present developments in the representation of vegetation dynamics in TRIFFID and include an evaluation of the expanded set of PFTs on simulated global vegetation distribution, and associated global carbon stocks and fluxes. This paper aims to demonstrate the overall performance of the new version of JULES in offline (not coupled to a climate model) simulations compared to both independent data sources and a previous version of the model.

## 2. Methods

### 2.1 JULES and TRIFFID

JULES simulates the processes of photosynthesis, autotrophic and heterotrophic respiration, and calculates the turbulent exchange of $CO_2$, heat, water, and momentum between the land surface and the atmosphere (Cox et al., 1998; Best et al., 2011; Clark et al., 2011). Vegetation dynamics are simulated by TRIFFID. Recently, new PFTs were added to JULES (Harper et al., 2016) (Table 1), which required updates to the TRIFFID competition scheme, described below. In this paper, we compare two versions of JULES: JULES-C1 and JULES-C2 based on JULES version 4.6. The former is a configuration of JULES with five PFTs as described in Harper et al. (2016) (called JULES5 in that paper) and as used in the TRENDY multi-DGVM synthesis project (Sitch et al., 2015). The latter (JULES-C2) is the new version, with 9PFTs and vegetation dynamics and updates described in Sections 2.2-2.3.

### 2.2 Vegetation dynamics and new height-based competition

Within TRIFFID, carbon acquired through NPP is allocated to either spreading (in other words increasing fractional coverage of a PFT in a grid cell) or growth (increasing height). The time evolution of fractional coverage of each PFT $i$ ($v_i$) is calculated as:

$$C_{V_i} \frac{dv_i}{dt} = \lambda_i \Pi_i v_* \left(1 - \sum_j c_{ij} v_j\right) - \gamma_{v_i} v_* C_{V_i} \tag{1}$$

where $C_V$ is the vegetation carbon (kgC m$^{-2}$), $\Pi$ is the accumulated NPP between calls to TRIFFID (kgC m$^{-2}$ (360 d)$^{-1}$), $v_*$ is the maximum of the actual fraction and a "seeding fraction" (0.01), and $\gamma_v$ is a PFT dependent parameter representing large-scale disturbance (360 d)$^{-1}$. In the present study, TRIFFID ran on a daily time step. The fraction of NPP allocated to spreading, $\lambda$, is a function of the balanced LAI, $L_{bal}$, which is the seasonal maximum of LAI based on allometric relationships (Cox, 2001):

$$\lambda = \begin{cases} 1 & for\ L_{bal} > L_{max} \\ \frac{L_{bal} - L_{min}}{L_{max} - L_{min}} & for\ L_{min} < L_{bal} \leq L_{max} \\ 0 & for\ L_{bal} \leq L_{min} \end{cases} \tag{2}$$

and the fraction allocated to growth is (1-$\lambda$). The PFT-dependent parameters $L_{max}$ and $L_{min}$ determine the balanced LAI at which plants allocate 100% of NPP toward expanding PFT coverage (spreading: $L_{bal} \geqq L_{max}$) or 100% toward vertical plant growth ($L_{bal} < L_{min}$).

Competition for space in the grid cell between PFT $i$ and the other PFTs is represented by the matrix $c_{ij}$, which represents a dominance hierarchy where height is the most important factor as it determines access to light. Effectively, the (1-$\Sigma c_{ij} v_j$) term in Eq. 1 is the space available to PFT $i$. In the original version of TRIFFID, trees were assumed to dominate shrubs, and shrubs were assumed to dominate grasses (Cox, 2001). Within tree (broadleaf and needle-leaf) and grass (C$_3$ and C$_4$) PFTs, there was co-competition and $c_{ij}$ was calculated as a function of vegetation height for the two competing PFTs:

$$c_{ij} = \frac{1}{1 + exp\left[20 * \frac{h_i - h_j}{h_i + h_j}\right]} \tag{3}$$

We made two changes to the original TRIFFID: first we removed the hard-wired dominance hierarchy (trees>shrubs>grasses) to allow for a generic number of PFTs. The dominancy hierarchy is now completely height-based, so that the tallest PFTs get the first opportunity to take up space in a grid cell. Second we removed co-competition, so that $c_{ij}$ is either 1 or 0. This simplifies the equilibrium solution for vegetation coverage (Section 3.2). When PFT $i$ is dominant, $c_{ij} = 0$ and PFT $i$ is not affected by PFT $j$; when type $j$ is dominant, $c_{ij} = 1$ and PFT $i$ does not have access to the space occupied by PFT $j$ ($v_j$).

**2.3 Updated parameters for JULES-C2**

Although the version of JULES described in this paper is similar to that described previously by Harper et al. (2016), there are four differences, which are summarized here. Impacts of the new equations for leaf, root, and stem nitrogen are discussed in detail in the Supplemental Material.

**2.3.1 Allometric parameters**

At the end of a TRIFFID timestep, the portion of NPP allocated toward growth increases the carbon content of leaves, roots, and wood. Both leaf and root carbon is linear with the balanced LAI, while total wood carbon ($C_{wood}$) is proportional to $L_{bal}$ based on the power law (Enquist et al., 1998):

$$C_{wood} = a_{wl} * L_{bal}^{b_{wl}} \tag{4}$$

The parameter $a_{wl}$ is a PFT-dependent coefficient relating wood to leaf carbon (units of kgC m$^{-2}$ per unit LAI), and $b_{wl}$ is a parameter equal to 5/3 (Cox, 2001). Previously, $a_{wl}$ was 0.65 for trees, 0.005 for grasses, and 0.10 for shrubs. After carbon pools are updated, canopy height is calculated from Eq. (5):

$$h = \frac{C_{wood}}{a_{ws}\eta_{sl}} * \left(\frac{a_{wl}}{C_{wood}}\right)^{1/b_{wl}} \tag{5}$$

The derivation of Eq. (5) is based on the assumption that total wood carbon is proportional to carbon in respiring stemwood ($S$), which itself is proportional to leaf area and canopy height ($h$)

based on the live stemwood coefficient, $\eta_{sl}$ (= 0.01 kgC m$^{-1}$ (m$^2$ leaf)$^{-1}$, derived from Friend et al. (1993)):

$$C_{wood} = a_{ws}S \tag{6}$$

$$S = \eta_{sl}h * L_b \tag{7}$$

In Eq. (6), $a_{ws}$ is the ratio of total wood carbon to respiring stem carbon, it was previously 10.0 for trees and shrubs and 1.0 for grasses, but this varies significantly with tree species: at least between 5 and 20 according to Friend et al. (1993). These ratios are relatively invariant with tree size and age within tree species or functional types, consistent with allometric relationships (e.g. Niklas and Spatz, 2004) and "pipe model" relationships between leaf-area and stem-area (e.g. Ogawa, 2015). As shown in the Results, there was a low vegetation carbon bias in JULES-C1, especially in regions dominated by broadleaf trees and shrubs. To increase vegetation carbon in areas where the model was lower than observed, we increased $a_{wl}$ and $a_{ws}$, while keeping their ratio constant, to the values given in Table 2. Changing $a_{wl}$ alone would affect the competitiveness of a PFT because it also affects plant height, $h$.

### 2.3.2 Soil respiration

JULES soil carbon is modelled with the Roth-C carbon model (Jenkinson, 1990; Coleman and Jenkinson, 2014). There are four pools: decomposable plant material (DPM), resistant plant material (RPM), microbial biomass (BIO), and humus (HUM). Respiration from each pool is calculated based on soil temperature ($T_{soil}$), moisture content ($s$), vegetation cover ($v$), and a pool-dependent turnover rate ($\kappa_i$):

$$R_i = \kappa_i * C_i * F_T(T_{soil}) * F_s(s) * F_v(v) \tag{8}$$

The turnover rates for the four soil carbon pools are 10 yr$^{-1}$ for DPM, 0.3 yr$^{-1}$ for RPM, 0.66 yr$^{-1}$ for microbial biomass, and 0.02 yr$^{-1}$ for humus (Coleman and Jenkinson, 2014). These are based on experiments on the decomposition of $^{14}$C labelled ryegrass over a 10-year period under field conditions (~9.3°C and > 20 mm of water) (Jenkinson, 1990). For both JULES-C1 and JULES-C2

in this paper, a $Q_{10}$ formulation was used for $F_T$ (Eq. 65 in Clark et al., 2011). However, only a fraction of respired carbon actually escapes to the atmosphere to represent the protective effect of small particles:

$$R_{soil \rightarrow atmos} = (1 - \beta_R) \sum_{i=1}^{scpool} R_i \qquad (9)$$

where

$$\beta_R = 1/[4.0895 + 2.672 * e^{-0.0786*Clayfrac}] \qquad (10)$$

Until version 4.6, JULES used a global clay fraction of 0.23 for this equation, which was based on the clay content at the site where the Roth-C model was calibrated. Now JULES uses a geographical variation of clay content based on the clay ancillary from the HadGEM2-ES CMIP5 simulations. All versions of the model presented in this study implement the global maps of clay.

### 2.3.3 Root and Stem Nitrogen

Third, new equations for root and stem nitrogen content ($N_{root}$ and $N_{stem}$, respectively) were added using updated data from the TRY database (Harper et al., 2016):

$$N_{root} = n_r * C_m * LMA * L_{bal} \qquad (11)$$

$$N_{stem} = \eta_{sl} * h * L_{bal} * n_{sw} \left[ \frac{1}{a_{ws}} + \left( 1 - \frac{1}{a_{ws}} \right) * hw_{sw} \right] \qquad (12)$$

where $C_m$ is the ratio of carbon per unit biomass (=0.4), LMA is the leaf mass per unit area for top of the canopy leaves, $n_r$ is the ratio of root N to root C, $n_{sw}$ is the ratio of stemwood N to stem C, and $hw_{sw}$ is the ratio of heartwood N to stemwood N. The latter is set to 0.5 based on a recommended range of 0.4-0.6 (Hillis, 1987). Parameters $n_r$ and $n_{sw}$ were calculated from the TRY database (Table 2).

### 2.3.4 Leaf nitrogen distribution

Fourth, updates were made to the parameter that characterizes the vertical distribution of leaf N through the canopy. Although these updates do not affect radiation interception through the canopy, they are referred to in the code as canopy radiation model 6 ("CRM6"). JULES splits the canopy

into 10 layers of equal LAI increment. In CRM6, leaf N declines exponentially through the canopy, so that for canopy layer $i$, the leaf N content ($N_{leaf}$, kg N m$^{-2}$) is:

$$N_{leaf_i} = N_m * LMA * e^{-k_{nl}*L_i} \tag{13}$$

where $N_m$ is leaf nitrogen per unit mass at the top of the canopy and $k_{nl}$ is a decay coefficient (=0.20). In JULES-C2 we update the value of $k_{nl}$ (Lloyd et al., 2010) and include the explicit term for LAI ($L$) in Eq. (13). The mean leaf N content is:

$$\overline{N_{leaf}} = \frac{N_m*LMA*(1-e^{-k_{nl}*L})}{k_{nl}*L} \tag{14}$$

Plant maintenance respiration is calculated as a function of the mean leaf nitrogen content. Impacts of the changes to leaf, root, and wood N are described in the supplementary material.

**2.4 Model evaluation**

The distribution of PFTs was evaluated by first dividing the land surface into eight biomes, based on the 14 World Wildlife Fund terrestrial ecoregions (Olson et al., 2001). The map of biomes (Fig. SM9) acted as a mask for the results to calculate biome-scale averages, and each grid cell was assumed to be 100% composed of the biomes shown in Fig. SM9. For each biome, we calculated the average fractional coverage of each PFT, average gridbox fluxes (GPP and NPP), and average gridbox carbon stocks (soils and vegetation), as well as average fractional coverage of agricultural land. These biome-scaled distributions and averages were then compared to observations. For observed PFT distribution, we used the global vegetation distribution from the European Space Agency's Land Cover Climate Change Initiative (ESA LCCCI) global vegetation distribution (Poulter et al., 2015; Hartley et al., 2017). To quantify the evaluation of PFT distribution, we calculated an error metric $\varepsilon$ for each PFT ($\varepsilon_i$ Eq. 15) and for each biome ($\varepsilon_B$ Eq. 16). The former enables a ranking of PFTs in terms of their improved distributions and is weighted by biome areas. The latter enables a comparison between models of the vegetation distribution on a biome scale and implicitly includes an area weighting since all fractions in a biome sum to 1.

$$\varepsilon_{i,PFT} = \sqrt{\frac{\sum_{B=1}^{8} A_B * (\nu_{B,i}^{mod} - \nu_{B,i}^{obs})^2}{\sum_{B=1}^{8} A_B}} \tag{15}$$

$$\varepsilon_{B,biome} = \sqrt{\frac{\sum_{i=1}^{npft} (\nu_{B,i}^{mod} - \nu_{B,i}^{obs})^2}{npft}} \tag{16}$$

In these equations, $A_B$ is the area of biome $B$, *npft* is the number PFTs (in this case 8 because C3

and C4 grasses are combined), and $\nu_{B,i}$ is the fractional coverage of PFT $i$ in biome $B$.

To evaluate the carbon fluxes, we used Gross primary productivity (GPP) from the Model Tree

Ensemble (MTE; Jung et al., 2011), and MODIS NPP from the MOD17 algorithm (Zhao et al.,

2005; Zhao and Running, 2010). Soil and vegetation carbon were from Carvalhais et al. (2014). In

addition, we compared biomass stocks to the data set from Ruesch and Gibbs (2008). In all

evaluations, we used model years corresponding to the available observation years: 1982-2011 for

GPP, 2000-2013 for NPP, and we used a 30-year period for soil and vegetation carbon (1980-2009).

All datasets were regridded to the model resolution of 1.25° latitude x 1.875° longitude.

**3. Model spin up and simulations**

**3.1 Model simulations**

There are a total of six simulations: one using JULES-C1 and five using JULES-C2. Both versions

of the model were run with transient climate, $CO_2$ and land use over the historical period. The

climate was from version 6 of CRUNCEP, which is a merged dataset of CRU and NCEP reanalysis

from 1901 to 2015. Climate variables used were downwelling longwave and shortwave radiation,

total precipitation, air temperature, specific humidity, zonal and meridional wind speeds, surface

pressure, and a constant diffuse fraction of shortwave radiation of 0.4. The fraction of agriculture in

each grid cell was included as fraction of crop and pasture from the harmonized dataset based on

HYDE3.2 (Hurtt et al., 2011). $CO_2$ concentration was from Dlugokencky and Tans (2013). We ran

three additional experiments with JULES-C2 to assess the contributions of climate change, land use

change (LUC), and $CO_2$ fertilization to the changes in carbon cycle components over the historical period (Table 5). Experiment $S_{CLIM}$ was forced with the transient climate from CRUNCEP-v6 to assess the contribution of climate change alone, while atmospheric $CO_2$ and land use were held to pre-industrial (1860) values. In experiment $S_{LUC,CLIM}$, climate and land-use changed, while $CO_2$ was held constant, and in experiment $S_{CO2,CLIM}$, climate and atmospheric $CO_2$ changed, while land-use was held constant. For the discussion of attributing changes to these drivers we refer to the main experiment as $S_{ALL}$, which has transient climate, LUC, and $CO_2$. The impact of LUC on the present-day carbon cycle is given by $S_{ALL}-S_{CO2,CLIM}$, and impact of $CO_2$ fertilization is given by $S_{ALL}-S_{LUC,CLIM}$. A fifth simulation with JULES-C2 was done to test the model with raw climate model output without bias correction to assess sensitivity of PFT distribution to the climate. This simulation was forced with the HadGEM2-ES RCP2.6 climate and $CO_2$. The available climate variables from HadGEM2-ES were downwelling longwave and shortwave radiation, stratiform rain, convective rain, stratiform snow, convective snow, air temperature, specific humidity, wind speed, surface air pressure, and the incoming diffuse shortwave radiation.

### 3.2 Estimating disturbance rates

The simulated distribution of PFTs in TRIFFID is sensitive to the large-scale disturbance parameter $\gamma_v$ from Eq. (1). The parameter represents several missing processes in JULES related to disturbance-induced mortality (such as fires, pests, and wind events), and provides an estimate of turnover rates for the PFTs. We developed a method for quickly estimating a global value of $\gamma_v$ for each PFT. Updated values of $\gamma_v$ were necessary due to new physiology, which resulted in a new NPP per PFT ($\Pi$ in Eq. 1), and an expanded set of PFTs. The method is based on a formula for the equilibrium distribution of PFTs, made possible by the removal of the hard-wired dominance hierarchy in TRIFFID. The equilibrium vegetation fractions are calculated by rearranging Eq. (1), meaning that for PFT $i$, the disturbance rate can be calculated as:

$$\gamma_{v_i} = \lambda_i \Pi_i \left[1 - \sum_{j=1}^{npft} c_{ij} v_j\right] * \frac{1}{c_{v_i}} \qquad (17)$$

where $n_{pft}$ is the number of PFTs.

To estimate new values for $\gamma_{vi}$, we ran JULES for 60 years under present-day climate, $CO_2$, and land-use, solving for the equilibrium vegetation fractions (as summarized in Section 7 of Clark et al., 2011). We used the simulated vegetation carbon ($C_v$), canopy height (to calculate the competition coefficients $c_{ij}$), and NPP for spreading ($\lambda\Pi$) at the end of the 60 years, together with the ESA LCCCI observed fraction of PFTs ($v_i$) (Poulter et al., 2015), to solve for $\gamma_{vi}$ in each grid cell. The result was a map of the $\gamma_v$ (~disturbance rate) per PFT required to get the observed PFT distribution based on simulated carbon available. Based on global distributions of $\gamma_v$ for each PFT in grid cells with <50% agriculture from 1950-2012, we used the median value in our simulations (Table 2). The new values of $\gamma_v$ do not guarantee a perfect simulation of PFT distribution, due to the use of one value per PFT, and because the calculation was based on solving the equilibrium solution to Eq. (1). However, this method does result in a range of $\gamma_v$ that make physical sense: there are low turnover rates for trees, high turnover rates for grasses, and moderate turnover rates for shrubs.

### 3.3 Spinning up vegetation and soil carbon

The vegetation fractions and soil carbon both require a long initial simulation to reach equilibrium. In a standard simulation, soil carbon spin-up needs to continue for 1,000-2,000 years after vegetation types have stabilized. There are two ways to speed this up: First by solving for vegetation fractions based on the equilibrium solution to Eq. (1); and second by using the 'modified accelerated decomposition' technique (modified-AD) (Koven et al., 2013). This results in a three-step spin up, summarized below. Note that the first two steps used CRUNCEP-v4, which was available at the beginning of the project.

   1) Solve for steady-state vegetation fractions in TRIFFID, increasing the time step for TRIFFID and phenology to 5 years and 10 days, respectively. Recycle the climate from the first 20 years of the simulation for a total of 60 years; in this case, CRUNCEP begins in

1900, so we recycled the 1901-1920 climate. In the simulations with HadGEM2-ES climate, the first 20 years of climate driving data is from 1860-1879. Specify land-use and $CO_2$ at their 1860 values.

2) Modified-AD: Run TRIFFID in dynamic mode with a time step of 1 day for TRIFFID and phenology using accelerated soil turnover rates (Table 3). Recycle climate from the first 20

years of the simulation for a total of 100 years. Initialize soil carbon to a global constant value of 3 kgC m$^{-2}$ to avoid any unrealistic values of soil carbon calculated during step 1. Specify land-use and $CO_2$ at their 1860 values.

3) Default decomposition: As above but use the default soil carbon turnover times. We initially used 200 years for this phase, but later in the project version 6 of the CRUNCEP climate

data became available, so the model was spun up an additional 200 years with the CRUNCEP-v6 data.

4) Begin the transient simulation from 1860, using transient $CO_2$, land-use, and climate. For CRUNCEP-v6, recycle the 1901-1920 climate for the first 41 years of the simulation.

In the last 100 years of the spin up, soil carbon changed by -0.06% and 0.43% with the CRUNCEP-v6 and HadGEM2-ES climates, respectively. These drifts are <6 PgC/100 years, or 2.8 ppm/100 years, which is below the C4MIP spin-up requirement for drifts of less than 10 ppm per century (Fig. SM7). Therefore, 300 years is adequate for spinning up the model, but there is a benefit to using 500 years: the drift in soil carbon in the CRUNCEP-v6 climate from years 200-299 was -3.5

PgC, compared to only -0.9 PgC from years 400-499.

## 4. Results

We analyse the results of JULES-C2 with the CRUNCEP-v6 climate against observations, and against two other models: JULES-C1 with CRUNCEP-v6 and JULES-C2 with HadGEM2-ES.

Globally, the HadGEM2-ES climate has higher precipitation and incoming shortwave radiation at

the surface, but lower specific humidity than the CRUNCEP-v6 climate. The average air temperature is similar until the 1960s, after which CRUNCEP-v6 is slightly warmer (Fig. SM8).

### 4.1 Predicted vegetation distribution

We evaluate the distribution of vegetation with two methods. First, to compare JULES-C1 and JULES-C2, we aggregated the 9 PFTs into the original 5. Figure 1 shows fractional coverage in each grid cell of the five vegetation types and bare soil for the models and the observations (BT=broadleaf trees, NT=needle-leaf trees, C3=C3 grasses, C4=C4 grasses, SH=shrubs). Second, we calculated fractional coverage of each PFT in eight biomes based on the WWF ecoregions (Fig.

2). The eight biomes are tropical forests (TF), extra-tropical mixed forests (MF), boreal forests (BF), tropical savannas (TS), temperate grasslands (TG), tundra (TU), Mediterranean woodland (Med), and deserts(D) (Figure SM9).

Most carbon in a tree/shrub is stored as woody biomass. Therefore, in terms of vegetation carbon

content, the most important distinction between plant types is between trees, grasses, and shrubs. With the CRUNCEP-v6 climate, JULES-C2 represents the distribution of these broad vegetation types very well (Fig. 1). There are several improvements compared to JULES-C1: for example, both the amount of tropical broadleaf trees in the central tropical forests and the spatial extent of boreal forests are more realistic in JULES-C2. The boreal forests in JULES-C1 do not extend far

enough across the North American and Eurasian continents. Instead, large areas of shrubs dominate at high latitudes. This bias is reduced in JULES-C2, although there is an underestimation (overestimation) in the coverage of needle-leaf trees in northeastern Eurasia (northern Europe).

Biome-scale distributions of the PFTs are shown in Figure 2, with results from JULES-C2 with

both the CRUNCEP-v6 and HadGEM2-ES climates. Differences between JULES-C2 run with different climates are typically small, with a tendency for the climate with higher precipitation to

result in more trees (Fig. 3) ($r^2 = 0.66$). Comparing the ESA vegetation fractions and CRUNCEP-v6 climate reveals a weaker positive relationship between tree coverage and annual rainfall ($r^2=0.36$). JULES is also sensitive to the specific humidity ($r^2=0.25$) but this is not supported by the ESA fractions. Coverage of needle-leaf deciduous trees ranges from 16% with the CRUNCEP-v6 climate to 27% with the HadGEM2-ES climate. This PFT was developed to have a competitive advantage in cold, dry environments. The annual average air temperature in the boreal forests is below freezing but precipitation is about 50% higher in the HadGEM2-ES climate compared to the CRUNCEP-v6 climate (Fig. SM8).

Agriculture is shown as a separate category since JULES can only grow C3 and C4 grasses in the agricultural fraction of grid cells. Agriculture accounts for 22-40% of all biomes except the two high latitude biomes (boreal forests and tundra). To compare with the ESA PFT distributions, we reduce the "observed" agricultural fraction (from the HYDE3.2 dataset) on grid cells where the prescribed agricultural fraction is greater than the coverage of ESA-observed grasses. This discrepancy between the observational datasets results in an apparent overestimation of agricultural fractions in some biomes. Although the agricultural fraction is prescribed, there can be bare soil on agricultural land if the JULES NPP is not sufficient to support vegetation (possibly due to the lack of irrigation in JULES). For this reason, in some biomes the agricultural fraction is underestimated (e.g. in temperate grasslands and deserts with JULES-C1).

JULES-C2 tends to overestimate the observed coverage of trees by 10-12% in tropical forests and savannahs, and by 3-5% in Mediterranean woodlands. The overestimation of trees in the tropical biomes is due to too much tropical broadleaf evergreen trees (BET-Tr). For example, in the tropical forest biome, 31% of the biome is covered with BET-Tr in the observations compared to a simulated range of 40-44% (with the HadGEM2-ES and CRUNCEP-v6 climates, respectively). The simulated coverage of broadleaf deciduous trees is very realistic in the tropical savannahs. The

coverage of dominant tree types is also close to observed in the boreal and mixed forests, with needle-leaf deciduous and evergreen trees in former and broadleaf deciduous and needle-leaf evergreen trees in the latter. However, the coverage of broadleaf deciduous trees is underestimated by 2-6% in both biomes.

Grasses are overestimated compared to observations by up to 21% in the boreal forests and tundra. The fractional coverage of bare soil is generally close to observed, with errors <5% for every biome except for tundra, where it is underestimated. In this biome, JULES-C2 produces 10-13% more shrubs and 10-21% more grass than observed. In the temperate grasslands, JULES-C2 with HadGEM2-ES climate overestimates the grass and needle-leaf evergreen tree coverage and underestimates bare soil coverage. Precipitation is almost twice as high in this biome in HadGEM2-ES compared to CRUNCEP-v6 (Fig. SM8). Shrubs in JULES-C2 tend to do best in cold environments: they are underestimated in tropical and mid-latitude biomes, very well simulated in the boreal forests, but overestimated in the tundra biome.

The total model biases based on bias per PFT are between 0.55-0.57 for all versions of the model (Table 4). The bias is an area-weighted fractional error per grid cell where the PFT exists (Eq. 15). The PFT biases are reduced for shrubs and grasses, but they are higher for broadleaf trees due to too many broadleaf trees in the tropics. The bias for needle-leaf trees in JULES-C2 depends on the climate: the bias is higher with the HadGEM2-ES climate compared to the CRUNCEP-v6 climate. Figure 2 also shows the bias calculated per biome for each simulation (Eq. 16). The biome biases are lowest in JULES-C2 with the HadGEM2-ES climate for five of the biomes, the exceptions being temperate grasslands, tundra, and deserts. In these biomes, the bias is lowest in JULES-C2 with the CRUNCEP-v6 climate. Comparing biomes, JULES-C2 represents vegetation distribution better in boreal and tropical forests than in mixed forests. The tropical savannahs have the highest bias.

**4.2 Terrestrial carbon cycle**

The patterns of gross and net primary production (GPP and NPP, respectively) simulated by JULES are similar to estimates derived from observations, although JULES fluxes are slightly high (Fig. 4). From 1982-2011, GPP is 133 PgC yr$^{-1}$ and 138 PgC yr$^{-1}$ according to JULES forced with CRUNCEP-v6 and HadGEM2-ES climate, respectively, compared to observation-based estimates from the same time period of 123±8 PgC yr$^{-1}$ (1982-2011; Beer et al., 2010). JULES-C1 with the CRUNCEP-v6 climate produces a higher GPP (143 PgC yr$^{-1}$). GPP is lower in JULES-C2 compared to JULES-C1, and closer to observations, in the tropical biomes (savannahs and forests, Fig. 5a).

From 2000-2013, MODIS estimates an NPP of ~55 PgC yr$^{-1}$, compared to 71 and 75 PgC yr$^{-1}$ in JULES with the CRUNCEP-v6 and HadGEM2-ES climates, respectively. During the same time period, JULES-C1 NPP is 66 PgC yr$^{-1}$. On average, NPP is 54% of GPP in JULES-C2, while it is 46% in JULES-C1. Both of these are similar to observation-based estimates that NPP should be roughly half of GPP. In JULES-C2, the largest overestimations of NPP occur in the tropical forests, savannahs, and mixed forests (Fig. 5b). JULES-C1 has high biases for GPP and NPP in tropical savannahs due to over-productive C4 grasses, and this bias is reduced in JULES-C2.

Global total vegetation carbon is 542 PgC and 553 PgC in JULES-C2 with the CRUNCEP-v6 and HadGEM2-ES climates, respectively, which is within the range supported by observations (400-600 PgC, Prentice et al., 2001), and is 65 PgC higher than the dataset from Ruesch and Gibbs (2008). The high bias mostly occurs in boreal and temperate forests and in tropical savannahs, where JULES produces more trees than observed (Fig. 5c). The spatial distribution of vegetation carbon is similar to observations (Fig. 4), but due to the extent of the broadleaf forests the total vegetation carbon in the tropical forest biome is higher than observed. However, there is large uncertainty in

global biomass datasets, for example the tropical savannah biome in JULES is very comparable to
the data from Carvalhais et al. (2014). JULES-C1 has lower vegetation carbon (468 PgC), with the
largest differences between the models being in the tropical forest and savannah biomes. There are
two reasons for the increase in $C_{veg}$ for JULES-C2. First, tropical evergreen and deciduous
broadleaf trees are more prevalent in JULES-C2 (Fig. 1). Second, the low vegetation carbon was
previously identified as a bias and the allometric parameters $a_{wl}$ and $a_{ws}$ were increased for
broadleaf trees (Section 2.3.1).

The largest biases in JULES occur for soil carbon, which is underestimated in both the high
latitudes and the tropics. Globally there is 1422 PgC in JULES-C2 with the CRUNCEP-v6 climate
and 1440 PgC with the HadGEM2-ES climate, compared to 2420 PgC in observations and 1362
PgC in JULES-C1. Soil carbon is the result of centuries (or longer) of litter accumulation. Woody
PFTs contribute more resistant material to the soil, while grasses turn over carbon in a more
decomposable form. Therefore, relatively small differences between simulations in PFT distribution
and NPP can contribute to large differences in the soil carbon. For example, in the tropics, soil
carbon is higher in JULES-C2 corresponding to the presence of more broadleaf trees and fewer
shrubs than in JULES-C1. In addition, due to the increased productivity simulated by JULES-C2,
the amount of carbon going into the soils through litterfall is also increased.

### 4.3 Transient carbon cycle

Over the past century and according to JULES-C2, the land surface was a net sink of carbon due to
an increase in soil carbon (+57 PgC) that offset a smaller decrease in vegetation carbon (-48 PgC)
(Fig. 6). The changes in brackets are the average during 2000-2009 minus average during 1900-
1909. These changes can be attributed to climate change acting on its own, climate change plus $CO_2$
fertilization, or climate change plus LUC. In the experiment with climate change only ($S_{CLIM}$, Table

5), vegetation carbon increases by 40 PgC, and there is a smaller increase in soil carbon since

warming encourages decomposition.

The effects of $CO_2$ fertilization and LUC on land carbon are given by the differences between

experiments $S_{ALL}$ and $S_{LUC,CLIM}$, and between $S_{ALL}$ and $S_{CO2,CLIM}$, respectively. Higher levels of $CO_2$

over the 20[th] century results in an additional 63 PgC of soil carbon and 49 PgC of vegetation

carbon. This is due to larger increases in NPP and litterfall than heterotrophic soil respiration ($R_h$).

Both NPP and $R_h$ are 58 PgC yr$^{-1}$ in 1900 in $S_{ALL}$. NPP increases to ~72 PgC yr$^{-1}$, while $R_h$

increases to 70 PgC yr$^{-1}$ by the end of the simulation. Land-use change results in a loss of 14 PgC of

soil carbon and 124 PgC of vegetation carbon. The largest reductions in vegetation carbon occur in

the tropics and in the eastern U.S. and Europe (Fig. 6). The total land-use source simulated by

JULES (138 PgC from 1900-2009) is very close to a recent estimate of total land-use and land

cover change emissions of 155±50 PgC from 1901-2012 (Li et al., 2017).

The annual sink is the net biosphere productivity (NBP), or NPP-$R_h$. The simulated NBP from

2000-2009 in JULES-C2 is 2.1±1.0 PgC yr$^{-1}$. The net land sink simulated by JULES is within the

range of estimates from both the Global Carbon Project (1.7±0.8 PgC yr$^{-1}$ over the same period, Le

Quéré et al., 2017) and the IPCC Fifth Assessment Report (AR5) (1.5±0.7 PgC yr$^{-1}$) (Table 6). The

JULES land sink is slightly high compared to the other two estimates, but this is not the case during

the 1980s and 1990s. Excluding LUC, JULES-C2 simulates an NBP of 3.4 PgC yr$^{-1}$ in the 2000s,

which is nearly double the natural NBP in the 1980s. The increase is due to a larger increase in

simulated NPP in the experiment without land-use change relative to the increase in $R_h$ (Fig. 6). In

$S_{ALL}$, the simulated NBP fluctuates around zero until the 1970s, after which it steadily increases due

to the fertilizing effect of atmospheric $CO_2$. Between 1980-2009, the NBP increases by 0.08 PgC yr$^{-}$

$^1$ yr$^{-1}$, due to a stronger positive trend in NPP (+0.27 PgC yr$^{-1}$ yr$^{-1}$) than in $R_h$ (+0.19 PgC yr$^{-1}$ yr$^{-1}$).

This increase is not seen in the experiment with preindustrial $CO_2$.

## 5. Discussion and Conclusion

Overall JULES with the new nine PFTs produces reasonable present-day distributions of vegetation, GPP, NPP, and vegetation carbon. The largest bias occurs for soil carbon, which is underestimated in regions where observations show a high soil carbon content – for example in peatlands and tundra. Global simulated GPP with JULES-C2 with observed climate is 133 PgC yr$^{-1}$, compared to GPP derived from up-scaled flux towers (123±8 PgC yr$^{-1}$; Beer et al., 2010) and GPP estimated from oxygen isotopes of atmospheric $CO_2$ (150-175 PgC yr$^{-1}$; Welp et al., 2011).

Global NPP according to MODIS is 55 PgC, consistent with another study that evaluated present-day NPP from 251 estimates in the literature and found a mean (±1 standard deviation) of 56.2 (±14.3) PgC yr$^{-1}$ (Ito, 2011). In comparison, the JULES NPP (71 PgC yr$^{-1}$) is slightly too high, which could be reduced by incorporating recent improvements to the parameterization of leaf dark respiration (Huntingford et al., 2017). JULES overestimates NPP in most biomes compared to MODIS, with the exception of deserts and temperate grasslands (Fig. 4). The highest overestimation of NPP is in the tropical forest biome, where JULES predicts a total NPP of 21.0 PgC yr$^{-1}$ compared to 15.4 PgC yr$^{-1}$ from MODIS. The MODIS algorithm estimates NPP using parameters derived from a DGVM (BIOME-BGC), climate, and satellite retrievals of land cover, fraction of absorbed photosynthetically available radiation (FPAR), and incoming radiation. Retrievals of reflectances like FPAR can saturate in regions with high vegetation density (Myneni et al., 2002; Lee et al., 2013), meaning that the tropical NPP from MODIS potentially has a low bias in tropical forests. Cloud contamination further complicates satellite retrievals of vegetation properties in the tropics (Cleveland et al., 2015). Future development and evaluation of carbon cycle models would greatly benefit from updated datasets of NPP that incorporate ground-based measurements from long-term networks and that provide uncertainty ranges. Regional products exist, for example the Global Ecosystems Monitoring (GEM) network

(http://gem.tropicalforests.ox.ac.uk/) and European National Forest Inventory (Neumann et al., 2016), which could be combined into a global dataset.

In a similar version of JULES with prescribed vegetation, simulated GPP and NPP were 128 and 62 PgC yr$^{-1}$, respectively (during the same time periods presented here) (Harper et al., 2016), compared to 133 and 71 PgC yr$^{-1}$, respectively, in this study. In that study, differences in PFT-level NPP did not affect the overall vegetation distribution owing to the prescribed distributions used. The simulations presented in the current study use dynamic vegetation, allowing JULES to predict

global vegetation distribution. Therefore, the productivity is slightly higher when JULES is allowed to predict vegetation distribution, although the previous study used older versions of CRUNCEP (v4) and JULES (v4.2 – see code availability).

JULES-C2 predicts global biomass of 542-554 PgC, with the largest high biases occurring in the

tropics and boreal forests. Early global estimates ranged from 400-600 PgC (Prentice et al. 2001), and the two datasets we analyzed estimate global biomass of 446-487 PgC. A more recent pan-tropical dataset of aboveground biomass suggests even lower vegetation carbon in the tropics (Avitabile et al., 2015). Despite the uncertainty in global biomass and NPP datasets, the fact that JULES overestimates both NPP and $C_{veg}$ in most biomes supports the conclusion that JULES net

productivity is too high. It's also possible that the allometric parameters $a_{wl}$ and $a_{ws}$ should be reduced following further evaluation of biomass predicted with the new PFTs. JULES tends to overestimate tree coverage and underestimate coverage by shrubs, which also contributes to high biomass. Woody trees dominate in regions where in reality shrubs form a larger proportion of the landscape, such as tropical savannahs and Mediterranean woodlands (Fig. 1, 2). In subtropical

forests, the model simulates too many broadleaf trees and virtually no shrubs.

Based on these evaluations, we highlight four priorities for developments of JULES vegetation: interactive fires, vegetation in semi-arid environments, impacts of soil moisture stress on vegetation, and tundra/high latitude vegetation. Interactive fires are an important missing process.

The simulation without land-use change (experiment $S_{CLIM,CO2}$) shows a large overestimation of biomass in the cerrado region of Brazil, where fires (in addition to human land clearing) likely limit vegetation coverage. Interactive fires could also help with the overestimation of trees and underestimation of shrubs, since shrubs occur earlier in the successional stages following a fire than trees. A lack of shrubs in tropical savannahs and Mediterranean woodlands also implies that future

development of PFTs should focus on vegetation characteristic of these biomes – for example drought-tolerant shrubs with phenology that responds to moisture as well as temperature. Such development should also take into account uncertainties in observed vegetation distributions in these regions (Hartley et al. 2017). The lack of vegetation in arid environments could also be due to plants experiencing too much moisture-related stress as soils dry, or to soils drying too rapidly

following a rain event. A revised parameterisation of soil moisture stress or more sophisticated vegetation hydraulics scheme would likely improve the model in these regions. Previous work also pointed to soil moisture stress as a likely culprit for underestimated dry season GPP at two towers in the Brazilian Amazon and for too low GPP at a non-irrigated maize site (Harper et al., 2016; Williams et al., 2017). Another large bias is the prevalence of shrubs in the tundra biome and

therefore more tundra-specific PFTs could improve the simulation in these regions. The importance of such developments should not be understated – climate change will likely bring a widening of subtropical dry zones and warmer temperatures at high latitudes, so these regions will be areas of large changes in vegetation in the future and will play key roles the evolving carbon cycle and ecosystem distribution of the 21[st] century.


JULES vegetation distribution and productivity fluxes seem robust to small differences in the climate based on the simulation with HadGEM2-ES climate, implying that different climate driving

datasets should not result in large differences in vegetation distribution. Global mean GPP, NPP, and $C_{veg}$ simulated with the two different climates varies by 5%, 7%, and <1%, respectively.

Vegetation distributions are broadly the same as well, although the extent of simulated trees is sensitive to precipitation. In contrast, simulated values of $C_{soil}$ have significant variation depending on the climate data used, since the soil carbon accumulates over centuries and is therefore sensitive to small differences in vegetation distribution and productivity. Global $C_{soil}$ is similar between the two simulations with JULES-C2, but the distribution has large regional differences (not shown). In

the case of soil carbon, the mismatch between simulated and observed is greater than the range between simulations.

Compared to the best available estimates of the annual terrestrial carbon sink, the JULES simulation is well within the range (2.0+1.0 PgC yr$^{-1}$ from 2000-2009). However, without nutrient limitation in

this version of the model, it's possible that the positive trend in NBP is too high in JULES, as indicated by the large simulated increase in NBP between the 1990s and 2000s in the experiment without land-use change, which is not found in the IPCC AR5 or GCP results. Although simulated NBP in the 1980s is bounded by the estimates from GCP and IPCC, the simulated NBP in the 2000s is higher than both constraints, indicating that either the increase in NPP is too large, or the

response from $R_h$ is too low. Anecdotally, the high bias in NPP (Fig. 4, 5) supports the former, but this doesn't rule out the possibility that respiration was undersensitive to climate and $CO_2$ over this period and the transient responses over the past 30 years should be further evaluated.

**Acknowledgements**

The authors acknowledge support from the Natural Environment Research Council (NERC) Joint Weather and Climate Research Programme through grant numbers NE/K016016/1 (A.B.H.) and NEC05816 (L.M.M.). NERC support was also provided to L.M.M. through the UK Earth System Modelling Project (UKESM, grant NE/N017951/1). A.B.H. also acknowledges support from her

EPSRC Fellowship (EP/N030141/1) and the EU H2020 project CRESCENDO (GA641816). The

EU project FP7 LUC4C (GA603542) provided support for S.S. & P.F. The Met Office authors were

supported by the Joint UK BEIS/Defra Met Office Hadley Centre Climate Programme (GA01101).

**Code availability**

This work was based on a version of JULES4.6 with some additional developments that will be

included in UKESM. The code is available from the JULES FCM repository:

https://code.metoffice.gov.uk/trac/jules (registration required). The version used was

r4546_UKESM (located in the repository at branches/dev/annaharper/r4546_UKESM). Two suites

are available to replicate the factorial experiments with CRUNCEP-v6 climate: u-ao199 and u-

ao216.

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

Table and Figure captions
**Table 1.** The original five and new nine PFTs in JULES.

**Table 2.** Updated parameters for vegetation carbon, and root and stem nitrogen in JULES-C2. The parameters are: $a_{wl}$ relates wood to leaf carbon (kg C m$^{-2}$ per unit LAI), $a_{ws}$ is the ratio of total wood carbon to respiring stem carbon, $n_r$ is the ratio of root N to root C, $n_{sw}$ is the ratio of stemwood N to stem C, $\gamma$ is the large-scale disturbance parameter (kg C m$^{-2}$ 360 d$^{-1}$).

**Table 3.** Turnover rates for the four soil carbon pools (RPM = resistant plant material; DPM = decomposable plant material; BIO = microbial biomass; HUM = humus). The factor is used to rescale soil carbon pools between the "fast" and "slow" spin ups.

**Table 4.** Bias in PFT distribution (Eq. 15) for JULES-C2 run calculated with two different climates and JULES-C1 run with the CRUNCEP-v6 climate.

**Table 5.** Simulated change in average fluxes and stocks from the period 1900-1909 to 2000-2009 in JULES-C2. Positive values indicate a gain of carbon by the land surface.

**Table 6.** Estimates of net land sink, emissions due to land-use change, and the "residual" sink on land from JULES compared to two other methods. Uncertainty ranges were reported differently for each method: for JULES ±1σ indicates the interannual variability of the annual mean, the IPCC reported a 90% confidence interval (based on Global Carbon Project 2013) which here is converted to ±1σ, and GCP reported ±1σ of the decadal mean across DGVMs for $S_{land}$ and ±1σ of bookkeeping estimates for $E_{LUC}$.

**Figure 1:** Fraction of land in each grid cell covered by vegetation and bare soil over the period 2010-2014 in the ESA LC-CCI dataset (left column), and JULES-C2 with CRUNCEP-v6 climate (middle column), and JULES-C1 with CRUNCEP-v6 climate (right column). BL = broadleaf; NL = needle-leaf.

**Figure 2:** Comparison of PFT distribution by biome in JULES-C2 forced with CRUNCEP-v6 and HadGEM2-ES climates, compared to JULES-C1 with CRUNCEP-v6 climate and to the observed distribution from ESA LC-CCI. The biomes are TF: Tropical Forests; MF: Temperate Mixed Forests; BF: Boreal Forests; TS: Tropical Savannah; TG: Temperate Grasslands; TU: Tundra; MED: Mediterranean Woodlands; D: Deserts. Biome distributions are shown in Fig. SM9. The black bars represent agricultural land. Model biases per biome are from Eq. (16).

**Figure 3:** Sensitivity of simulated tree coverage in each biome to precipitation, air temperature, specific humidity, and shortwave radiation. Model results are from JULES with both CRUNCEP-v6 and HadGEM2-ES climates. The observations compare the ESA LC-CCI land cover to the observed (CRUNCEP-v6) climate.

**Figure 4**: Simulated and observed GPP, NPP, vegetation and soil carbon. Results are shown from JULES-C2 and JULES-C1 both with CRUNCEP-v6 climate. Sources for observations are: GPP: FLUXNET-derived model tree ensemble (Jung et al., 2011); NPP: MODIS17 (Zhao et al., 2005); $C_{veg}$: Ruesch and Gibbs (2008); $C_{soil}$: Carvalhais et al. (2014).

**Figure 5:** Biome-averaged (a) GPP, (b) NPP, (c) $C_{veg}$, and (d) $C_{soil}$ in JULES-C1 and JULES-C2 (both with CRUNCEP-v6 climate) compared to observations. The observation sources are the same as in Fig. 4 except (c) compares the $C_{veg}$ from Ruesch and Gibbs (2008) ("RG08") to that from Carvalhais et al. (2014) ("C14", black shapes). The biomes are TF: Tropical Forests; MF: Temperate Mixed Forests; BF: Boreal Forests; TS: Tropical Savannah; TG: Temperate Grasslands;

TU: Tundra; MED: Mediterranean Woodlands; D: Deserts (biomes in Fig. SM9). Grid cells with >50% agriculture have been excluded from the biome averages.

**Figure 6:** Global mean gross primary productivity (GPP), net primary productivity (NPP), heterotrophic respiration ($R_{het}$), net biome productivity (NBP = GPP-$R_{het}$), vegetation carbon ($C_{veg}$), and soil carbon ($C_{soil}$). Global means are shown for the $S_{CLIM,LUC}$, $S_{CLIM,CO2}$, and $S_{ALL}$ experiments summarized in Table 5.

**Figure 7:** Global distribution of vegetation carbon in JULES-C2 in experiments (average from 2000-2009) with and without transient land-use and $CO_2$ based on the experiments summarized in Table 5.

| 5 PFTs (JULES-C1) | 9 PFTs (JULES-C2) |
| --- | --- |
| Broadleaf trees (BT) | Tropical broadleaf evergreen trees (BET-Tr) |
| Needle-leaf trees (NT) | Temperate broadleaf evergreen trees (BET-Te) |
| C3 grass (C3) | Broadleaf deciduous trees (BDT) |
| C4 grass (C4) | Needle-leaf evergreen trees (NET) |
| Shrubs (SH) | Needle-leaf deciduous trees (NDT) |
|  | C3 grass (C3) |
|  | C4 grass (C4) |
|  | Evergreen shrubs (ESH) |
|  | Deciduous shrubs (DSH) |

**Table 1.** The original five and new nine PFTs in JULES.

|           | BET-Tr  | BET-Te  | BDT     | NET     | NDT     | C3 grass | C4 grass | ESH     | DSH     |
|-----------|---------|---------|---------|---------|---------|----------|----------|---------|---------|
| $a_{wl}$  | 0.845   | 0.78    | 0.78    | 0.65    | 0.80    | 0.005    | 0.005    | 0.13    | 0.13    |
| $a_{ws}$  | 13      | 12      | 12      | 10      | 10      | 1        | 1        | 13      | 13      |
| $n_{sw}$  | 0.0072  | 0.0072  | 0.0072  | 0.0083  | 0.0083  | 0.01604  | 0.0202   | 0.0072  | 0.0072  |
| $n_r$     | 0.01726 | 0.01726 | 0.01726 | 0.00784 | 0.00784 | 0.0162   | 0.0084   | 0.01726 | 0.01726 |
| $\gamma$ initial | 0.005 | 0.005 | 0.005 | 0.007 | 0.007 | 0.20 | 0.20 | 0.05 | 0.05 |
| $\gamma$ from Eq. 17 | 0.007 | 0.014 | 0.007 | 0.020 | 0.010 | 0.25 | 0.06 | 0.10 | 0.06 |

**Table 2.** Updated parameters for vegetation carbon, root and stem nitrogen in JULES-C2. The parameters are: $a_{wl}$ relates wood to leaf carbon (kg C m$^{-2}$ per unit LAI), $a_{ws}$ is the ratio of total wood carbon to respiring stem carbon, $n_r$ is the ratio of root N to root C, $n_{sw}$ is the ratio of stemwood N to stem C, $\gamma$ is the large-scale disturbance parameter (kg C m$^{-2}$ 360 d$^{-1}$).

|                        | RPM              | DPM             | BIO              | HUM               |
|------------------------|------------------|-----------------|------------------|-------------------|
| Default ($s^{-1}$)     | $3.17 \times 10^{-7}$ | $9.6 \times 10^{-9}$ | $2.1 \times 10^{-8}$ | $6.4 \times 10^{-10}$ |
| Accelerated ($s^{-1}$) | $3.17 \times 10^{-7}$ | $3.17 \times 10^{-7}$ | $3.15 \times 10^{-7}$ | $3.2 \times 10^{-7}$ |
| Factor                 | 1                | 33              | 15               | 500               |

**Table 3.** Turnover rates for the four soil carbon pools (RPM = resistant plant material; DPM = decomposable plant material; BIO = microbial biomass; HUM = humus). The factor is used to rescale soil carbon pools between the "fast" and "slow" spin ups.

| PFT | JULES-C2 CRUNCEP-v6 | JULES-C2 HadGEM2 | JULESC1-CRUNCEP-v6 |
|---|---|---|---|
| **Bet-Tr** | 0.15 | 0.14 | 0.13 (for all BT) |
| **BET-Te** | 0.017 | 0.015 | -- |
| **BDT** | 0.063 | 0.049 | -- |
| **NET** | 0.078 | 0.12 | 0.15 (for all NT) |
| **NDT** | 0.043 | 0.044 | -- |
| **Grasses** | 0.088 | 0.096 | 0.11 |
| **ESH** | 0.053 | 0.054 | 0.17 (for all Shrubs) |
| **DSH** | 0.054 | 0.056 | -- |
| **Total bias** | 0.55 | 0.57 | 0.56 |

**Table 4.** Bias in PFT distribution for JULES-C2 run with two different climates and JULES-C1 run with the CRUNCEP-v6 climate.

| | JULES-C2 ($S_{CLIM}$) | JULES-C2 ($S_{ALL}$) | JULES-C2 ($S_{CLIM,LUC}$) | JULES-C2 ($S_{CLIM,CO2}$) |
|---|---|---|---|---|
| **Experiment summary** | Transient climate change only | Transient $CO_2$, land-use, and climate change | Transient climate and land-use change | Transient climate and $CO_2$ with 1860 land-use |
| **$\Delta C_{soil}$ (PgC)** | 8 | 57 | -6 | 71 |
| **$\Delta C_{veg}$ (PgC)** | 40 | -48 | -97 | 75 |

**Table 5.** Simulated change in average fluxes and stocks from the period 1900-1909 to 2000-2009 in JULES-C2. Positive values indicate a gain of carbon by the land surface.

|  | 1980-1989 | 1990-1999 | 2000-2009 |
|---|---|---|---|
| ***Net land sink*** | | | |
| JULES-C2 (NBP in $S_{ALL}$) | 0.5±1.1 | 1.1±0.8 | 2.1±1.0 |
| IPCC AR5 | 0.1±0.6 | 1.1±0.7 | 1.5±0.7 |
| GCP 2017 ($S_{land}$-$E_{LUC}$) | 0.7±0.7 | 1.2±0.5 | 1.7±0.8 |
| ***Emissions from LUC*** JULES-C2 ($NBP, S_{CLIM,CO2}$-$S3_{ALL}$) | -1.2±1.1 | -1.3±0.9 | -1.3±1.0 |
| IPCC AR5: net LUC[1] | -1.4±0.6 | -1.5±0.6 | -1.1±0.6 |
| GCP 2017 ($E_{LUC}$)[2] | -1.2±0.7 | -1.3±0.7 | -1.2±0.7 |
| ***Residual Land sink*** | | | |
| JULES-C2 (NBP in $S_{CLIM,CO2}$) | 1.7±1.1 | 2.4±0.9 | 3.4±1.0 |
| IPCC AR5 | 1.5±0.8 | 2.6±0.9 | 2.6±0.9 |
| GCP 2017 ($S_{land}$) | 2.0±0.6 | 2.5±0.5 | 2.9±0.8 |

[1] Using the bookkeeping LUC flux accounting model of Houghton et al. (2012).
[2] Bookkeeping methods

**Table 6.** Estimates of net land sink, emissions due to land-use change, and the "residual" sink on land from JULES compared to two other methods. Uncertainty ranges were reported differently for each method: for JULES ±1σ indicates the interannual variability of the annual mean, the IPCC reported a 90% confidence interval (based on GCP 2013) which here is converted to ±1σ, and GCP reported ±1σ of the decadal mean across DGVMs for $S_{land}$ and ±1σ of bookkeeping estimates for $E_{LUC}$.

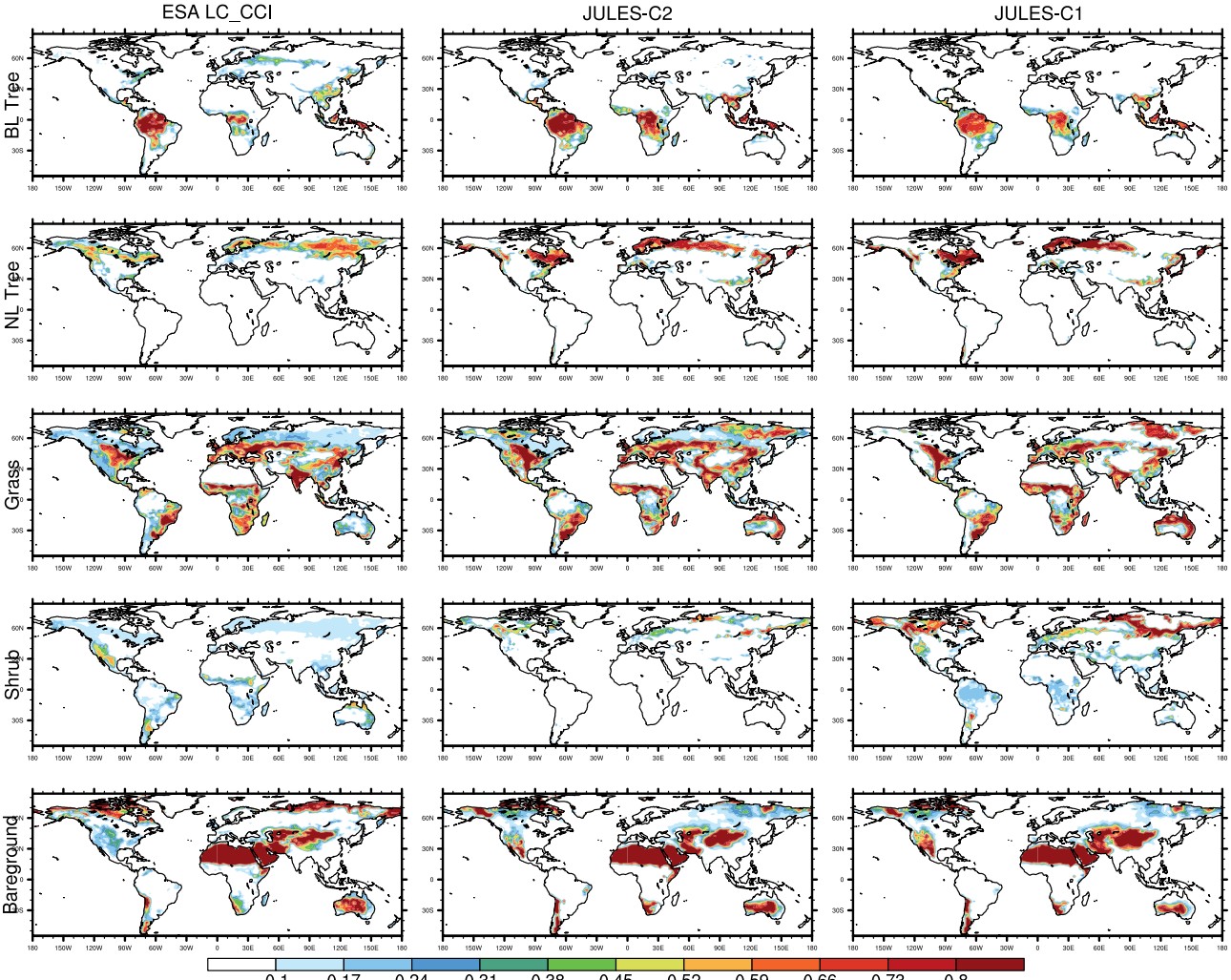

Figure 1

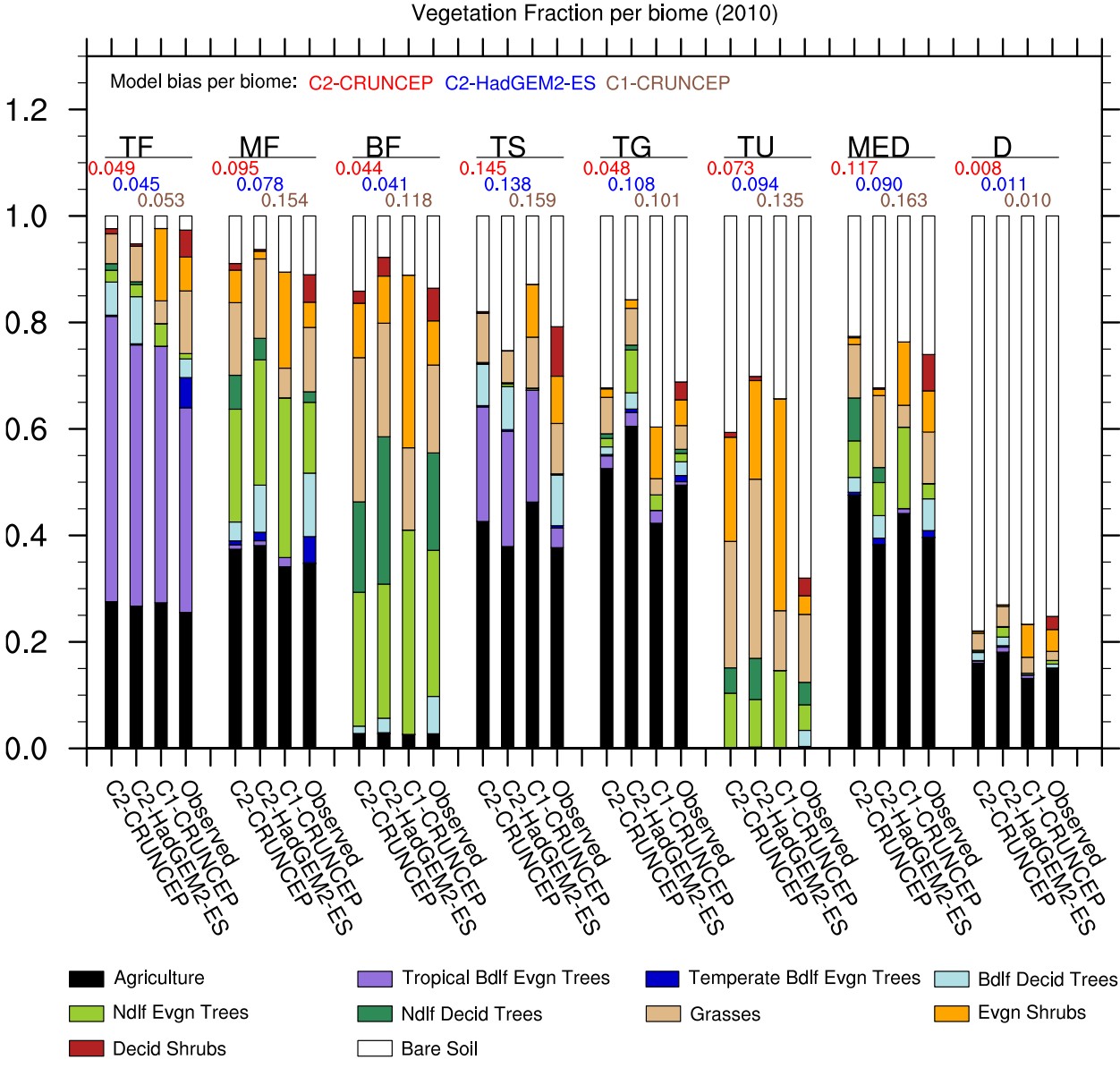

Figure 2

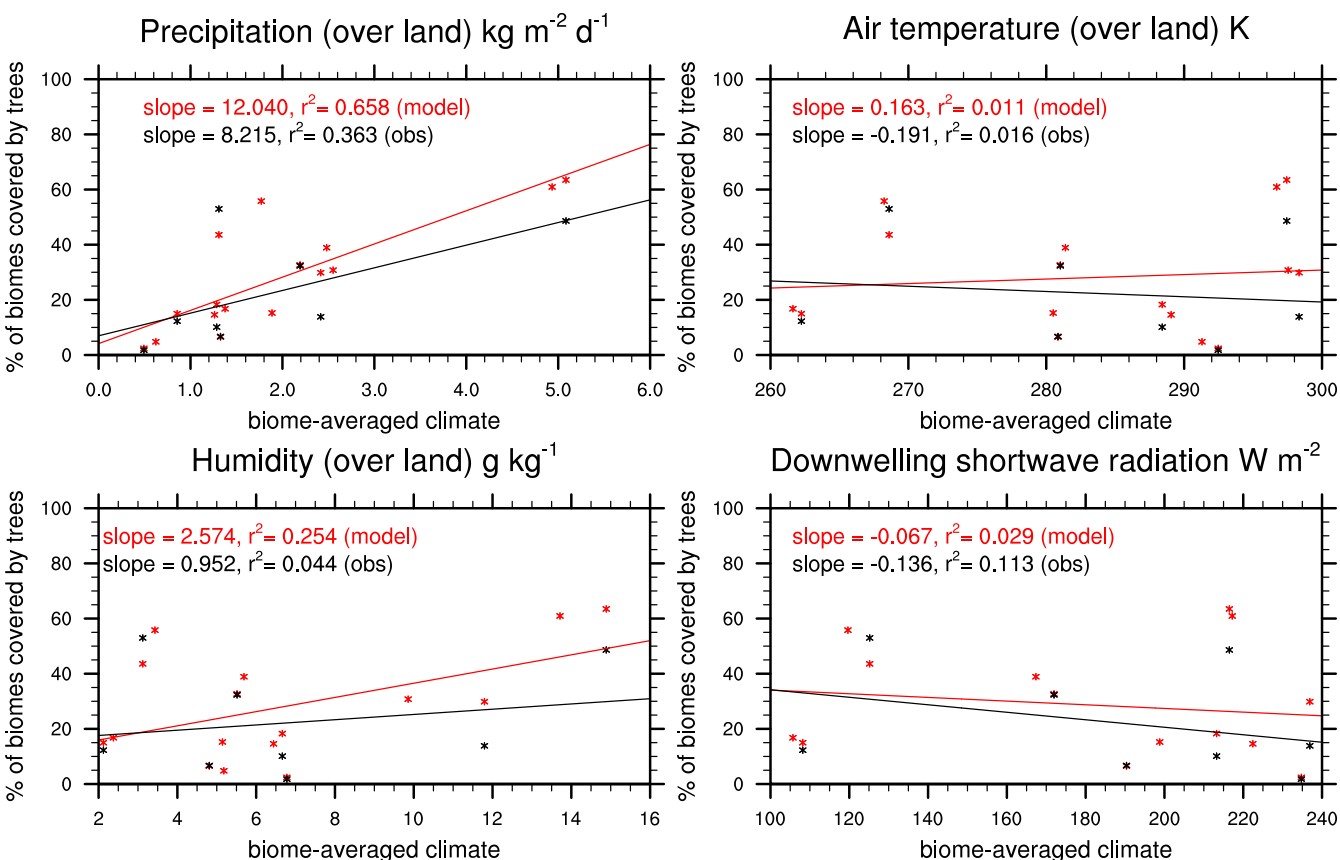

Figure 3

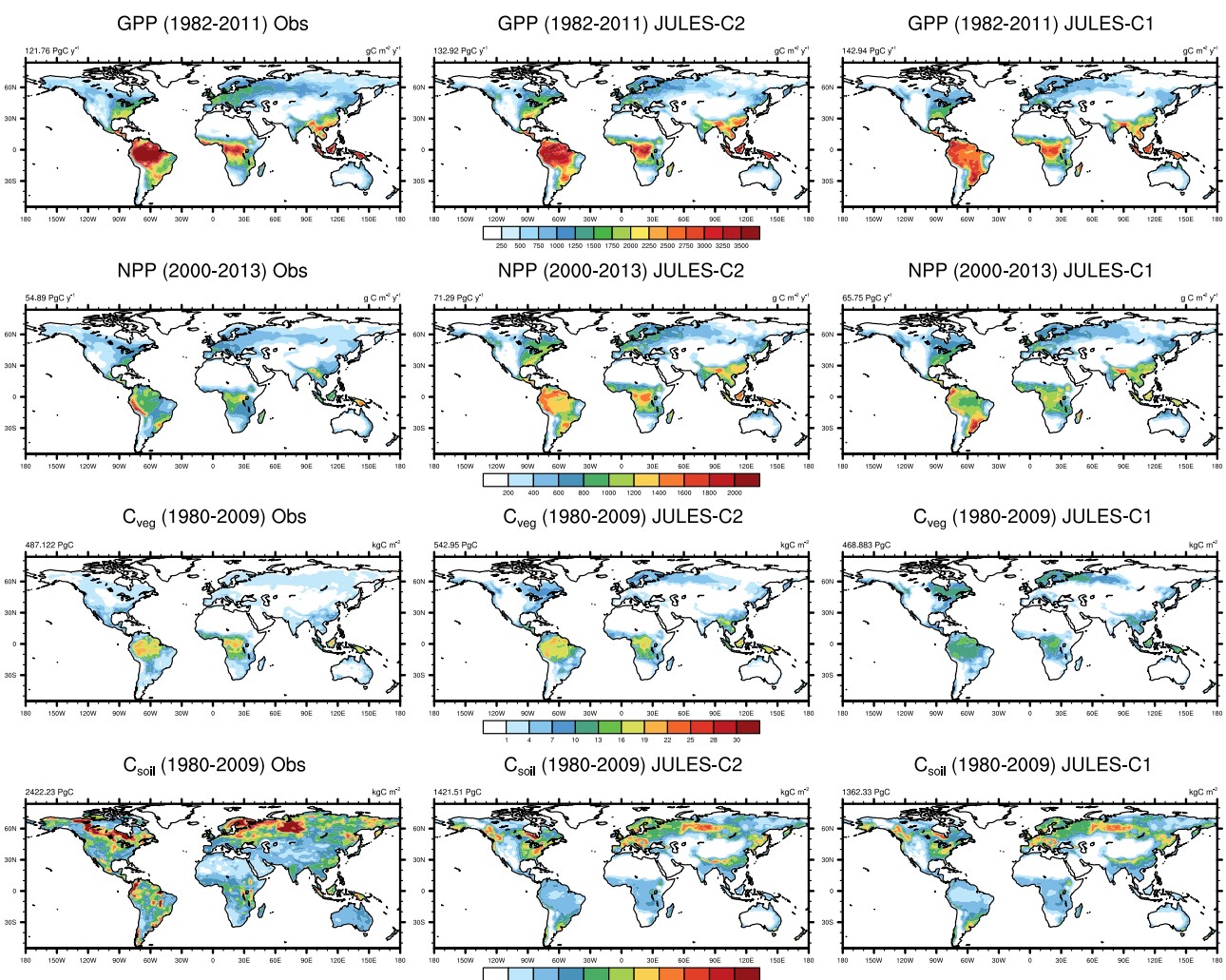

Figure 4

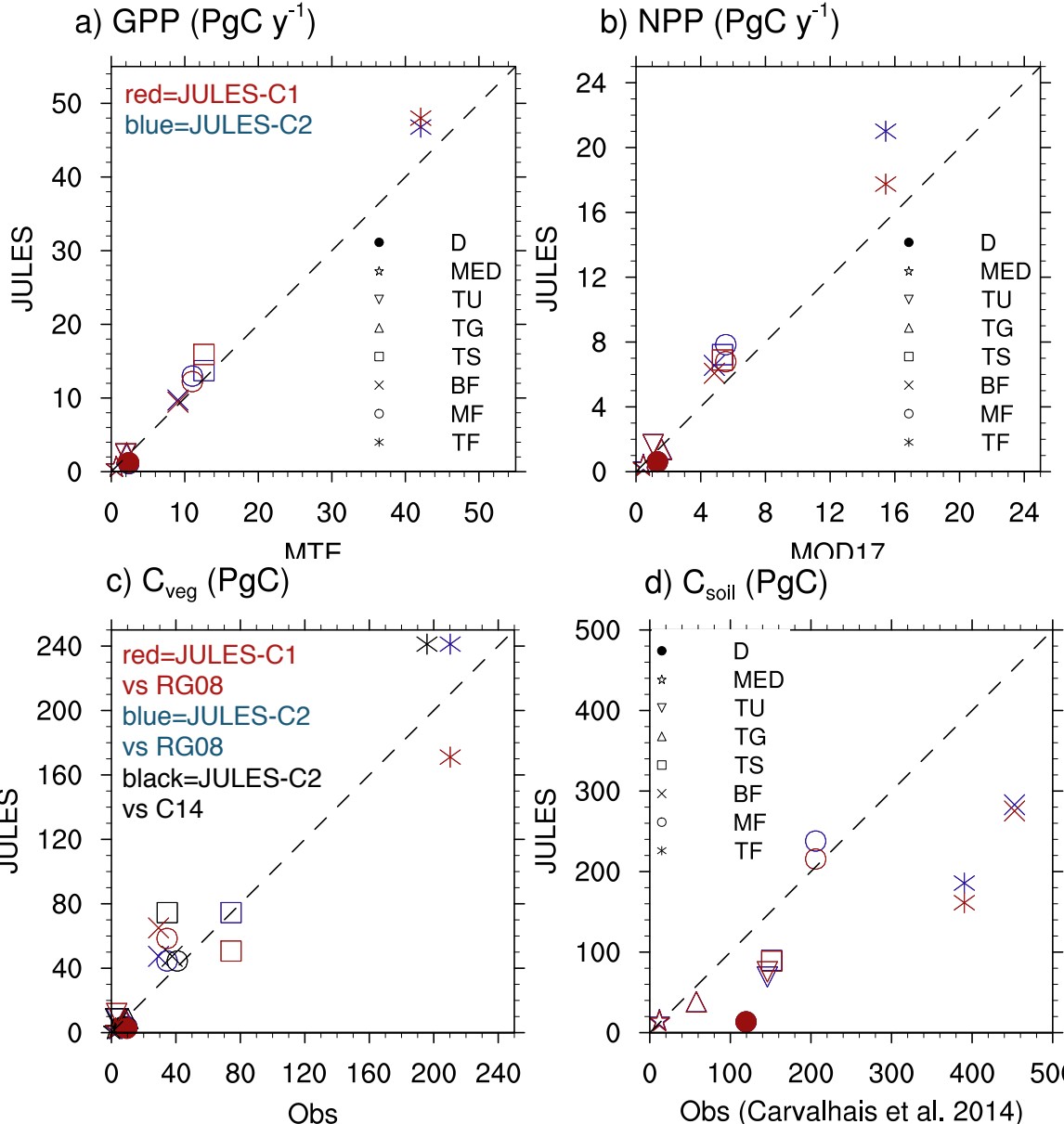

Figure 5

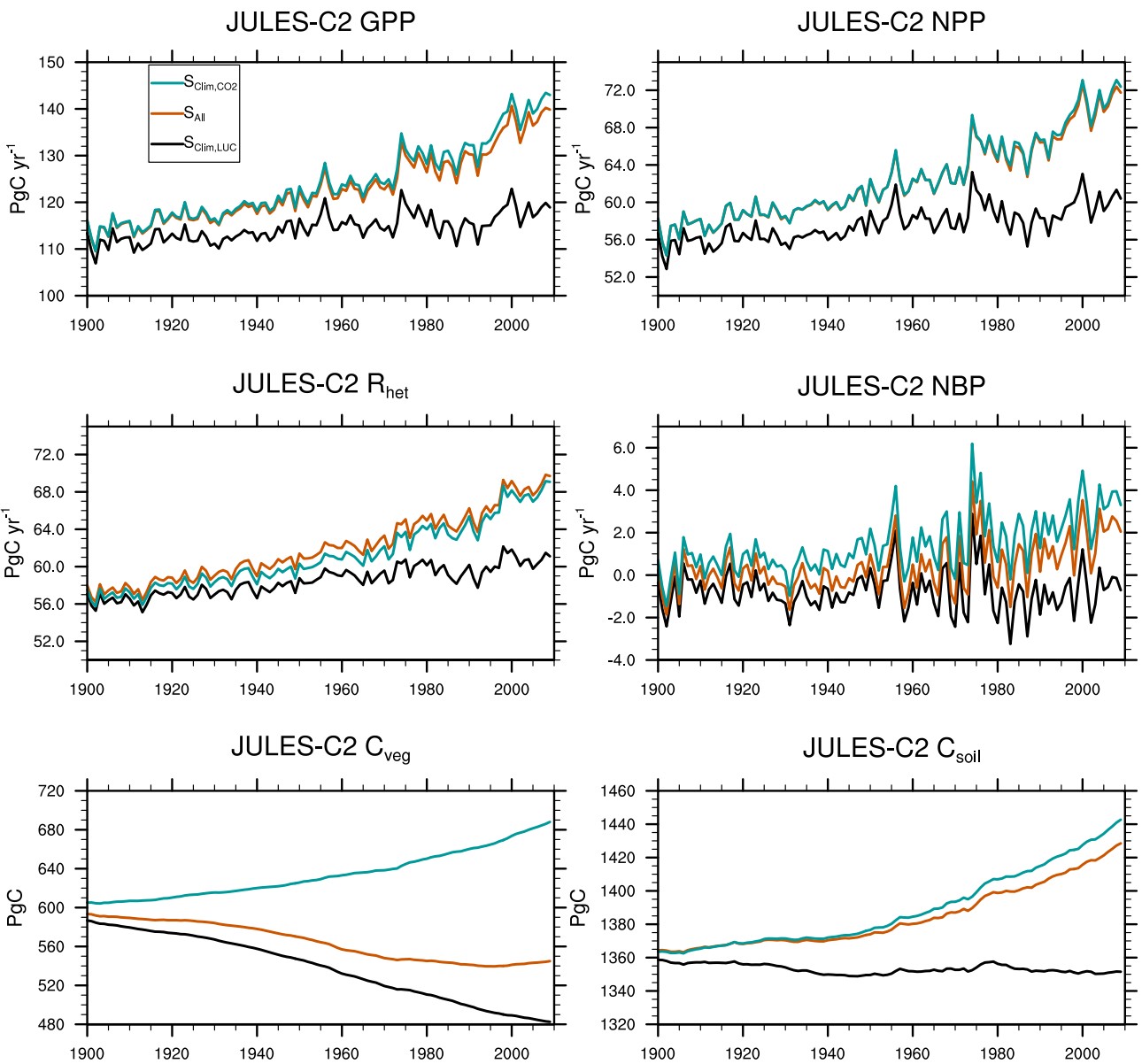

Figure 6

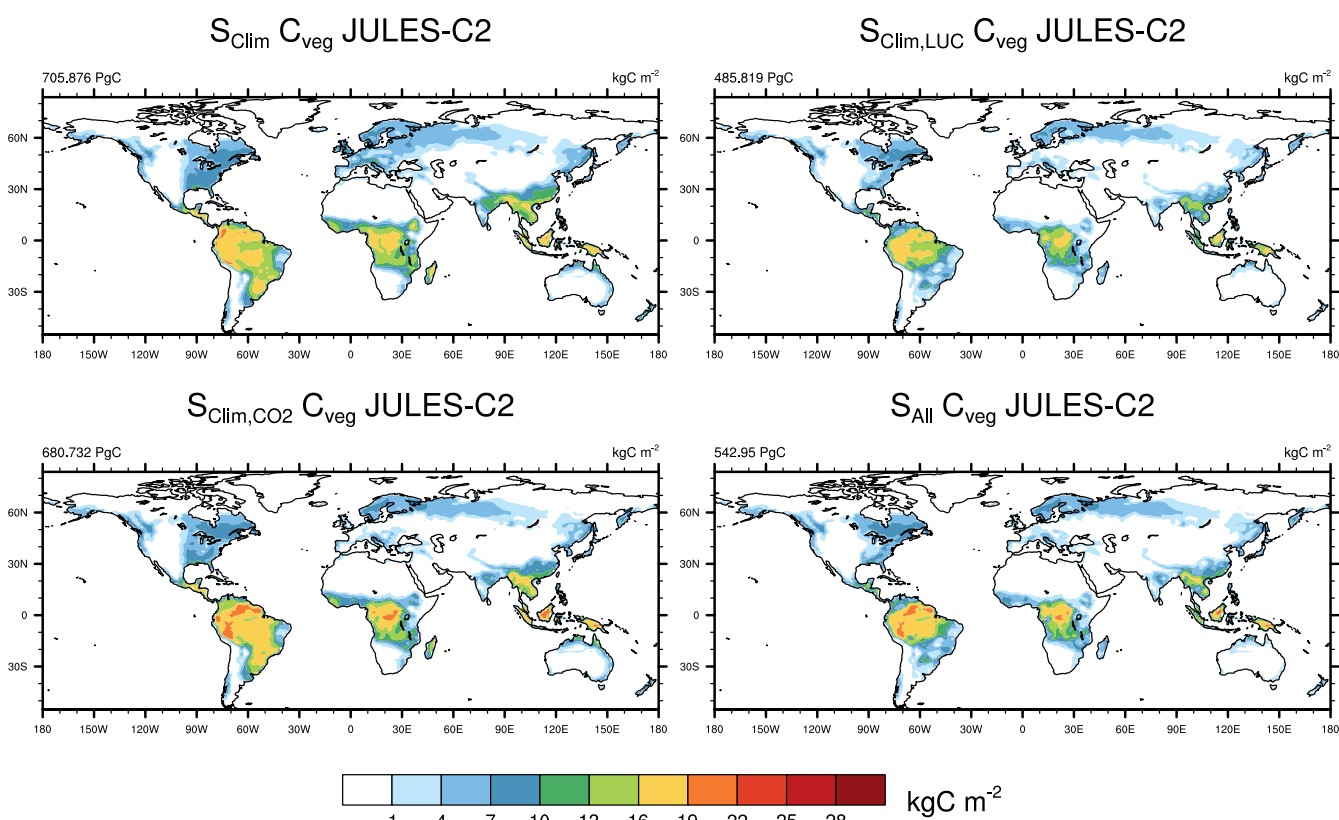

Figure 7