# Peer review of "Vegetation distribution and terrestrial carbon cycle in a carbon-cycle configuration of JULES4.6 with new plant functional types"

_Geoscientific Model Development, 2017_

## Referee Comment (RC1) · V. Haverd (Referee) · 19 Feb 2018

This manuscript presents changes to the TRIFFID vegetation dynamics module of the JULES land surface model. In general I found the manuscript to represent a significant advance, with good documentation , clear evaluation and a summary of how the new configuration of JULES simulates historical terrestrial carbon balance.

I recommend the manuscript as a valuable contribution to GMD, subject to a few comments below being addressed:

1. Line 160-165. What are the observational rages for aw1 and aws. Are the adjusted

[Figure]

values within observed ranges? How applicable are fixed values of these parameters for stands of different ages, and what are the implications for carbon accumulation in young vs old-growth forests?

2. Biomass from Ruesch and Gibbs (2008). Could you justify why you don't use more up-to-date biomass products, eg GEOCARBON (Avitabile et al. 2016), https://www.wur.nl/en/Expertise-Services/Chair-groups/Environmental-Sciences/Laboratory-of-Geo-information-Science-and-Remote-Sensing/Research/Integrated-land-monitoring/Forest_Biomass.htm

3. Figure 1. What quantity is being mapped ? Is it area fraction? Please make it clear in the text and figure caption.

4. L340. If the agricultural land is prescribed, how can the model under-estimate it?

5. L. 435. "in agreement with the high bias in simulated NPP". Please revise this stated agreement. High bias in NPP doesn't necessarily give high NBP. It is the magnitude NPP, relative to that of heterotrophic respiration that dictates the magnitude of the land carbon sink, with the difference related to the different rates of change of these two fluxes.

6. Figure SM7: different colours or line-styles are needed to distinguish the two simulations.

7. Can you explain why biomass in central Africa is well simulated (Fig. 3-4), but vegetation distribution is not (Fig 1).

8. Satellite-based NPP. Please consider referencing uncertainty in this product, for example, satellite-based NPP datasets have large uncertainties in tropical regions (Cleveland et al. 2015), e.g. from saturation of the fraction of photosynthetically active radiation (FPAR) in high vegetation density areas.

---

## Referee Comment (RC2) · Anonymous Referee #2 · 28 Feb 2018

This paper presents a new version of the JULES land surface model with an increased number of plant functional types and some important changes to its vegetation dynamics parameterizations and to its nitrogen cycle. The model output is essentially evaluated on the global scale, partly per biomes. The paper is well written and clear. It is short enough to be a not too painful read, yet it contains all the essential information it requires to serve its purpose, that is, to be used a reference paper for this and probably future versions of JULES using the same vegetation dynamics parameterizations. I have only a few minor suggestions for this paper.

- Line 148: Please refer to Table 2 already here

[Figure]

- Line 259: Could be written more clearly. Equilibrium vegetation mean dv/dt = 0 in eq (1), which directly yields eq. 17. Please use Sum symbol in eq. 17 (as done in eq 16), no need to keep eq 16 which is almost identical.

- To this point, it should be stressed that the disturbance parameters are tuned such that the equilibrium vegetation is OK. Therefore the relatively good results in fig 1 and 2 should not be too surprising. Please state this clearly also in the discussion. Are these disturbance parameters realistic? Any chance to evaluate them?

- Line 327: Small differences between the results with CRUNCEP and HadGEM-ES climate. Is this due to the fact the the HadGEM-ES climate is so realistic, or does this suggest that JULES is not very sensitive to climate differences for some reason, except for some specific PFTs mentioned explicitly?

- Figure 2: Hard to evaluate the differences between the different model configurations with the naked eye. Could one add an error statistic for each of the biomes and model configs? You could consider adding the PFT names (that is, spelling out their acronyms) in the legend, might be helpful.

- Carbon spinup: The carbon spinup figure in the supplement shows that some PFTs are quite far from equilibrium at the end of the accelerated spinup, and that large instantaneous adjustments follow after that, which seem to have consequences over the whole transient period. Please discuss potential impacts on the soil carbon evaluation and discuss briefly.

---

## Editor Comment (EC1) · D.A. Ham (Editor) · 14 Jun 2018

Late in the review process it was noticed that some of the authors on this manuscript are from the same institution, the UK Met Office, as the handling topical editor. Although this is a situation we would usually attempt to avoid, its discovery so late in the process means that the conflict would not effectively be removed by changing editor, and to do so would damage the transparency of the editorial process. Instead, I have audited the editorial process. I can confirm that the reviewers, who are the primary independent part of the peer review process, are both highly qualified and completely independent. In particular, they are not in any way associated with the JULES model

but work on different terrestrial models. Readers can therefore be assured that the peer review process for this manuscript has been robust.

---

## Author Response (AR1)

We thank the reviewers for their helpful comments on the manuscript. Below we address their individual questions and comments and note where revisions occur in the marked-up version of the manuscript. Our responses are in blue, with the updates to the manuscript italicized. The relevant line numbers in the revised manuscript are provided in red.

In the marked-up manuscript, blue text denotes additions and deletions of text, and green text denotes text that was moved.

**Reviewer 1**
1. Line 160-165. What are the observational rages for aw1 and aws. Are the adjusted values within observed ranges? How applicable are fixed values of these parameters for stands of different ages, and what are the implications for carbon accumulation in young vs old-growth forests?

Response:
The ratio of live sapwood/stem to wood (parameter $a_{ws}$) varies significantly with tree species: at least between 0.05 and 0.28 according to Friend et al. (1993), which corresponds to $a_{ws}$ between 4 and 20. Our values of $a_{ws}$ (10-13 for woody vegetation) are therefore well within the range of observations. Although we expect variations in $a_{ws}$ and $a_{wl}$ across tree species, these are ratios that are relatively invariant with tree size/age within tree species or functional types, consistent with allometric relationships (e.g. Niklas & Spatz, 2004) and "pipe model" relationships between leaf-area and stem-area (e.g. Ogawa, 2015).

*We will add this context into the Methods section where the new values of $a_{ws}$ and $a_{wl}$ are given.* This has been added, see lines 160-163.

2. Biomass from Ruesch and Gibbs (2008). Could you justify why you don't use more up-to-date biomass products, eg GEOCARBON (Avitabile et al. 2016), https://www.wur.nl/en/Expertise-Services/Chair- groups/Environmental-Sciences/Laboratory-of-Geo-information-Science-and-Remote- Sensing/Research/Integrated-land-monitoring/Forest_Biomass.htm

Response:
The Ruesch and Gibbs (2008) biomass is an estimate of total carbon in above- and below-ground biomass, while GEOCARBON only covers aboveground biomass in forests. Therefore, Ruesch and Gibbs (2008) allows a more thorough comparison of the JULES outputs. However there is large variation in estimates of biomass, as was pointed out in Avitabile et al. (2016). We therefore also compared our biomass estimates to a dataset from Carvalhais et al. (2014) (Fig. 3c), which were derived using Saatchi et al. (2011) for the Tropics and Thurner et al. (2014) for boreal and temperate regions.

*We will include some further context in the discussion addressing the uncertainty in global datasets of biomass.* See lines 550-555.

3. Figure 1. What quantity is being mapped ? Is it area fraction? Please make it clear in the text and figure caption.

Response:
Figure 1 shows the fraction of vegetation types and bare soil in each grid cell. *We will update the text and the figure caption.* This has been changed.

4. L340. If the agricultural land is prescribed, how can the model under-estimate it?

Response: The fraction of land in a grid cell dedicated to agriculture is specified based on Hyde data set. We assume that only C3 and C4 grasses can grow on agricultural land, but if the simulated NPP is not sufficient to support vegetation, there will be bare soil instead. This is why the agricultural fraction can be underestimated. To compare with the ESA PFT distribution, we reduced the "observed" agricultural fraction on grid cells where the prescribed agricultural fraction is greater than the grasses in the grid cell. This is why the agricultural fraction can also be higher than indicated by ESA.

*We will clarify these two points in the text.* See Lines 385-392.

5. L. 435. "in agreement with the high bias in simulated NPP". Please revise this stated agreement. High bias in NPP doesn't necessarily give high NBP. It is the magnitude NPP, relative to that of heterotrophic respiration that dictates the magnitude of the land carbon sink, with the difference related to the different rates of change of these two fluxes.

Response: This is a good point. There are two points we should make here. First, the doubling of NBP was due to an increase in NPP not matched by an increase in heterotrophic respiration. Second, although simulated NBP in the 1980s was in between estimates from GCP and IPCC, the simulated NBP in the 2000s was higher than both constraints, indicating that either the increase in NPP was too large, or the response from Rh was too low. The high bias in NPP noted earlier in the paper supports the former, but this doesn't rule out the possibility that Rh was undersensitive to climate and $CO_2$ over the period 1980-2009.

*We will update the text with this information.* See Lines 503-504 and 614-619.

6. Figure SM7: different colours or line-styles are needed to distinguish the two simulations.

*Response: We will add colors to the lines in this Figure.* Figure has been updated, also in updating the figure we found that an additional 200 years of spin up was performed with the CRUNCEPv6 climate, which we have added to SM Figure 7 (years 100-299 in the new figure were missing in the previous version). The rate of change of soil carbon is smoother with all of the data included (see response to final comment from Reviewer 2).

7. Can you explain why biomass in central Africa is well simulated (Fig. 3-4), but vegetation distribution is not (Fig 1).

Response: In Central Africa, the model simulates too many broadleaf trees and virtually no shrubs, although these are regions with large uncertainties in the ESA vegetation observations (Hartley et al. 2017). A similar bias is apparent in subtropical South America. In both regions (outside of the central tropical forests), fires suppress tree growth and this process is missing in the model. The overestimation of trees and underestimation of shrubs is also apparent in Figure 2. The spatial distribution of vegetation carbon is similar to observations (Fig. 3), but due to the extent of the broadleaf forests the total vegetation carbon in the Tropical Forest biome is higher than observed (Fig. 4).

*We already mention the fact that this version of JULES does not have interactive fires (new lines 577-579), so we will add to this discussion as these are regions where we expect an improvement after representing the effects of fires.* See lines 454-456, 557-560.

8. Satellite-based NPP. Please consider referencing uncertainty in this product, for ex- ample, satellite-based NPP datasets have large uncertainties in tropical regions (Cleve- land et al. 2015), e.g. from saturation of the fraction of photosynthetically active radiation (FPAR) in high vegetation density areas.
Response: There are a few papers estimating the uncertainty in MODIS NPP in the Tropics, but unfortunately the product does not contain uncertainty estimates. There is approximately a 15% uncertainty in global estimates of NPP (Ito 2011). Also as the reviewer mentioned, there is large disagreement in NPP from inventories, satellites, and models in the tropics (Cleveland et al. 2015).

*We will add some discussion in the paragraph describing NPP results (beginning at the top of page 19) about the uncertainty in this product to give some context to the apparently high NPP in JULES.* See Lines 520-538.

**Reviewer 2**
- Line 148: Please refer to Table 2 already here.

Response:
We will correct this in the revised manuscript.

- Line 259: Could be written more clearly. Equilibrium vegetation mean dv/dt = 0 in eq (1), which directly yields eq. 17. Please use Sum symbol in eq. 17 (as done in eq 16), no need to keep eq 16 which is almost identical.

- To this point, it should be stressed that the disturbance parameters are tuned such that the equilibrium vegetation is OK. Therefore the relatively good results in fig 1 and 2 should not be too surprising. Please state this clearly also in the discussion. Are these disturbance parameters realistic? Any chance to evaluate them?

In response to the above two points:
*We agree that Eq. 16 is not necessary, and we will correct Eq. 17. Also, we will update this text to make it clearer, and to better explain the process for selecting new values of the disturbance parameter.*

Equation 17 was used to calculate a new $\gamma$ for each PFT in each gridcell. We chose the new parameter value as the median value in grid cells with <50% agriculture (Lines 270-273). The simulated distribution of PFTs would be even more realistic if we used spatially varying values for $\gamma$, but part of the reason for the model not recreating observed distributions is because of the use of one global value per PFT. Also the calibration of $\gamma$ was based on a simulation in equilibrium, while Figures 1 and 2 show distributions after the transient run, when vegetation fractions are not in equilibrium.

*We will highlight these points in the revised text.*

*We will also update this text to better describe what the disturbance parameters are representing:* The parameter represents several missing processes in JULES related to disturbance-induced mortality (such as fires, pests, and windthrow). The value of the parameter would be difficult to evaluate since it represents several processes, although the general values give an estimate of turnover rates for the PFTs. As such, the numbers do make sense: there are low turnover rates for trees, high turnover rates for grasses, and moderate rates for shrubs.

The above points are addressed in the updated text in Section 3.2, particularly Lines 282-287, 294-301.

- Line 327: Small differences between the results with CRUNCEP and HadGEM-ES climate. Is this due to the fact the the HadGEM-ES climate is so realistic, or does this suggest that JULES is not very sensitive to climate differences for some reason, except for some specific PFTs mentioned explicitly?

*Response: We plan to add a figure to the SM comparing the temperature and precipitation of the CRUNCEP and HadGEM2-ES climates.*

The HadGEM2-ES does broadly represent the present climate very well, but there are biases in the model that have an impact on the vegetation distribution. Mapping the climates will enable us to link differences in simulated vegetation fraction to differences in the two driving climates.
We have added Figure 3 showing the relationship between simulated and observed fractions of tree coverage per biome and biome-average climate variables (temperature, precipitation, specific humidity and shortwave radiation). Also see new text at Lines 344-346, 370-379, 415-416. We also added supplemental Figure SM8 to show precipitation, temperature and humidity in each biome.

- Figure 2: Hard to evaluate the differences between the different model configura- tions with the naked eye. Could one add an error statistic for each of the biomes and model configs? You could consider adding the PFT names (that is, spelling out their acronyms) in the legend, might be helpful.

Response: Thank you for this suggestion – *we will include the full names of the PFTs in the legend.* Also, the error metric was calculated per PFT (Eq. 15), but it could just as easily be calculated per biome. *We will do this and add that information to the plot.*
We have updated Figure 2 and added the equation for the bias per biome (Eq. 16). There is new text, see lines 229-236 for description of the new statistic and 425-430 for the results.

- Carbon spinup: The carbon spinup figure in the supplement shows that some PFTs are quite far from equilibrium at the end of the accelerated spinup, and that large in- stantaneous adjustments follow after that, which seem to have consequences over the whole transient period. Please discuss potential impacts on the soil carbon evaluation and discuss briefly. First we point out that the figure is showing soil carbon in the four soil pools, not PFTs. But the reviewer is right in pointing out some questionable behaviour in this figure. *We will double-check the plotting routine for errors.* The turnover times for the RPM, biomass, and humus pools were increased during the accelerated decomposition phase. To create this figure, the soil carbon pools were multiplied by the rescale factors for each pool (Table 3),

but it's possible we did not apply the correct scaling factors. The pool sizes should be smaller during the accelerated decomposition phase, instead of being larger as they are in the RPM and Biomass pool for CRUNCEP.

We found that an additional 200 years of spin up was performed with the CRUNCEPv6 climate, which we have added to SM Figure 7 (years 100-299 in the new figure were missing in the previous version). The rate of change of soil carbon is smoother with all of the data included, except for the RPM pool which decreased by ~20 PgC at the end of the accelerated phase. The decrease occurred with both climate forcings, and we are confident it's not a mistake in the plotting. The RPM pool readjusted quickly, and it changed by at most 1 PgC over the last 100 years of the spin up. The figure has been updated in the SM, and we have clarified these points in the new manuscript (Section 3.3, particularly lines 334-339).

[revised manuscript text omitted]

---

## Author Response (AR2)

Dr Anna B Harper
Department of Mathematics
University of Exeter
Exeter, UK
Email: a.harper@exeter.ac.uk

22 June 2018

Dear Sir or Madam,

We are submitting a final version of our manuscript, *Vegetation distribution and terrestrial carbon cycle in a carbon-cycle configuration of JULES4.6 with new plant functional types*, for publication in Geoscientific Model Development. The only change since our previous submission is that we added information to the acknowledgements and updated some figure captions. We also will upload separate pdf files for each figure. Please let me know if you require anything else for the final version.

Yours Faithfully,

Dr Anna Harper (on behalf of my co-authors)